# Influence of Group-IVA Doping on Electronic and Optical Properties of ZnS Monolayer: A First-Principles Study

**DOI:** 10.3390/nano12213898

**Published:** 2022-11-04

**Authors:** Bin Liu, Wan-Sheng Su, Bi-Ru Wu

**Affiliations:** 1School of Mathematics and Physics, Nanyang Institute of Technology, Nanyang 473004, China; 2National Taiwan Science Education Center, Taipei 11165, Taiwan; 3Department of Electro-Optical Engineering, National Taipei University of Technology, Taipei 10608, Taiwan; 4Department of Physics, National Sun Yat-sen University, Kaohsiung 80424, Taiwan; 5Division of Natural Science, Center for General Education, Chang Gung University, Tao-Yuan 33302, Taiwan

**Keywords:** the ZnS monolayer, doping, electronic property, first-principles

## Abstract

Element doping is a universal way to improve the electronic and optical properties of two-dimensional (2D) materials. Here, we investigate the influence of group−ⅣA element (C, Si, Ge, Sn, and Pb) doping on the electronic and optical properties of the ZnS monolayer with a tetragonal phase by using first-principles calculations. The results indicate that the doping atoms tend to form tetrahedral structures with neighboring S atoms. In these doped models, the formation energies are all negative, indicating that the formation processes of the doped models will release energy. The formation energy is smallest for C−doped ZnS and gradually increases with the metallicity of the doping element. The doped ZnS monolayer retains a direct band gap, with this band gap changing little in other element doping cases. Moreover, intermediate states are observed that are induced by the sp^3^ hybridization from the doping atoms and S atoms. Such intermediate states expand the optical absorption range into the visible spectrum. Our findings provide an in-depth understanding of the electronic and optical properties of the ZnS monolayer and the associated doping structures, which is helpful for application in optoelectronic devices.

## 1. Introduction

Since the discovery of graphene [1], more and more two-dimensional (2D) layered materials have been discovered [2,3,4,5,6]. Owing to their distinct electronic, optical, and thermoelectric performance, many studies have been devoted to boosting the practical applications of 2D materials [7,8,9]. Among these 2D materials, binary Ⅱ-Ⅵ zinc chalcogenides exhibit the potential for broad applications in optoelectronic and thermoelectric devices [10,11,12,13].

Bulk ZnS presents a cubic zinc-blende (ZB) structure at low temperatures, with a band gap of 3.77 eV [14]. The stable 2D phase of ZnS is a honeycomb structure, analogous to graphene [15,16,17]. Previous investigations have revealed that graphene-like 2D ZnS possesses an ultrawide band gap of 4.3 eV [18] and transparency larger than the bulk phase in the visible range [19]. To make the most of the ZnS monolayer, it was rolled into a single-walled nanotube to improve its electronic performance [20]. Moreover, strain modulation and element doping in the ZnS monolayer were also carried out [18,21,22]. Similar to many 2D materials with more than one allotrope [4,23,24], the graphene-like ZnS monolayer also has an allotrope [25], which exhibits a tetragonal phase just like monolayer CdTe [26], CdSe [27], CdS [27], and ZnSe [28]. In our previous study, we have explored the influence of strain on the electronic and optical properties of the tetragonal ZnS [29], but a more complete understanding of monolayer tetragonal ZnS is still insufficient [30]. 

Zn atoms form tetrahedrons with neighboring S atoms in tetragonal ZnS, and group−ⅣA elements also tend to form tetrahedrons when connecting with the chalcogens. Therefore, when one Zn atom in the ZnS monolayer is substituted with group−ⅣA elements, the doped ZnS will retain its stable structure but exhibit some distinct electronic and optical properties. For this purpose, in this work, we have substituted one Zn atom in the ZnS monolayer supercell with group−ⅣA elements and then studied the electronic and optical properties of the doped monolayer tetragonal ZnS using *ab initio* calculations. The results indicate that the doping elements also form tetrahedral structures resembling the Zn atom in the doped ZnS, which could ensure the structural stability of the doped models. The formation energies are negative, indicating that these doping models will release energy during the formation processes, with formation energy decreasing with the increasing metallicity of the doping element. The doped ZnS monolayer retains the direct band gap, and the values of the band gaps change little, except for C−ZnS. Moreover, intermediate states are found in the band gap, which leads to better optical absorption in the visible spectrum. Our results reveal the influence of group−ⅣA element doping on the electronic and optical properties of the ZnS monolayer, which is helpful for its application in optoelectronic devices.

## 2. Computational Details

The Vienna ab initio simulation package (VASP) based on the density functional theory (DFT) was utilized to carry out the first-principles calculations [31,32]. The projector augmented-wave (PAW) method [33] and Perdew–Burke–Ernzerhof generalized gradient approximation (GGA–PBE) were utilized to describe the exchange–correlation functional [34]. The kinetic energy cutoff was set to 500 eV, and k-point meshes of 10 × 10 × 1 were chosen from the Brillouin zone with the Monkhorst–Pack method [35]. Based on the primitive cell of the ZnS monolayer with the tetragonal phase in our previous study [29], a 3 × 3 supercell was built, composed of 18 Zn and 18 S atoms, with a vacuum thickness of 20 Å. By replacing one Zn atom in the supercell with the group-ⅣA element (C, Si, Ge, Sn, and Pb), the doped ZnS models were obtained with a doping fraction of 2.8% (i.e., C−ZnS, Si−ZnS, Ge−ZnS, Sn−ZnS, and Pb−ZnS). To further optimize the doped structure models, the convergence values of 10^−6^ eV/atom for total energy and 0.01 eV/Å for atomic force component were applied. Finally, the electronic and optical properties were calculated.

## 3. Results and Discussion

Figure 1 shows the supercell structure of the ZnS monolayer, and the doped models in which the central Zn atom in the ZnS monolayer are substituted with C, Si, Ge, Sn, and Pb atoms, respectively, in C−ZnS, Si−ZnS, Ge−ZnS, Sn−ZnS, and Pb−ZnS. In the ZnS monolayer without doping, the atoms are uniformly distributed in the supercell. However, the models become distorted after the element doping owing to the mismatch of the atomic size. From Figure 1, the structure of doped ZnS is obviously different depending on the doping atom. Notably, the C doping shrinks the local structure, whereas the Si, Ge, Sn, and Pb doping expand the local structure. To evaluate the influence of the doping atom on the monolayer structure, the interatomic distances in the models are studied. The central Zn atom in the ZnS monolayer is represented as the *O* atom, with the *A*, *B*, *C*, and *D* atoms corresponding to the first, second, third, and fourth nearest atoms to the *O* atom, as shown in the ZnS monolayer in Figure 1. In the doped models, the *O* atom is replaced by the doping atoms of C, Si, Ge, Sn, and Pb, respectively, and the interatomic distance between *O* atom and the *A*, *B*, *C*, and *D* atoms can reflect the extent of structural variation.

Figure 2 shows the interatomic distance of *OA*, *OB*, *OC*, and *OD* varying with the element doping. For the nearest neighboring *A* atom, the length of *OA* in C−ZnS becomes shorter, while it gradually increases in Si−ZnS, Ge−ZnS, Sn−ZnS, and Pb−ZnS. For the second, third, and fourth neighboring atoms, the lengths of *OB*, *OC*, and *OD* decrease slightly in C−ZnS, while those in Si−ZnS, Ge−ZnS, Sn−ZnS, and Pb−ZnS are lengthen, showing a similar trend to the first atomic shell. Compared to the variations in *OB*, *OC*, and *OD*, the changes in *OA* are more significant; the changes in *OB*, *OC*, and *OD* are small and shrink as the atoms lie farther from the central *O* atom. This indicates that the influence of elemental doping is mainly concentrated in the closest neighboring atoms. 

To identify the influence of doping on the structural stability, the formation energy, denoted as *E*_form_, is calculated by the following equation: *E*_form_ = *E*_doped_ + *E*_Zn_ − *E*_ZnS_ − *E*_ⅣA_(1)

*E*_doped_ and *E*_ZnS_ are the total energies of the doped ZnS and pure ZnS without doping, respectively. The values of *E*_Zn_ and *E*_ⅣA_ are energies of the isolated atoms (Zn and group−ⅣA: C, Si, Ge, Sn, and Pb) in the cubic cells with an edge length of 20 Å, respectively. The computed *E*_form_ values of C−ZnS, Si−ZnS, Ge−ZnS, Sn−ZnS, and Pb−ZnS are −1.82, −1.54, −1.27, −0.18, and −0.05 eV, respectively, as seen in Figure 3a. Note that the formation energies of the doped models are negative, suggesting that the doped models may have a higher stability owing to their smaller energy. The formation energy of C−ZnS is the smallest, indicating that the largest energy will be released when substituting the Zn atom with the C atom in monolayer tetragonal ZnS. Notably, the formation energy decreases gradually with the increase in the metallicity of doping atoms. The formation energy of Pb−ZnS is largest, suggesting that Pb doping into the ZnS monolayer is relatively difficult. 

Figure 3b shows the Bader charge of the doped atoms and the associated first nearest neighbor atoms in the doping models, which can be used to evaluate the charge transfer in the doping structures. Usually, there are four and six electrons in the outermost shell of the group-IVA elements and the S atom, respectively. From Figure 3b, the Bader charge of the C atom is 4.1, while each S atom in C−ZnS is 6.6. This suggests that the C atom obtains its charge from the bonding S atoms and the S atoms obtain their charge from the bonding Zn atoms. However, the Bader charges of Si, Ge, Sn, and Pb are ~3.2 for Si−ZnS, Ge−ZnS, Sn−ZnS, and Pb−ZnS, and those of S atoms are ~6.8. This indicates that the Si, Ge, Sn, and Pb atoms will lose charge to the bonding S atoms. 

To explore the influence of element doping on the electronic structure of the ZnS monolayer, the band structures of the ZnS monolayer, C−ZnS, Si−ZnS, Ge−ZnS, Sn−ZnS, and Pb−ZnS are calculated, as displayed in Figure 4. In line with the previous studies [29,30], the ZnS monolayer has a direct band gap of 2.91 eV, in which both the valence band maximum (VBM) and conduction band minimum (CBM) are located at the Γ point. When one Zn atom is replaced by C, Si, Ge, Sn, and Pb atoms, respectively, both the VBM and CBM of these doped models are still located at the Γ point, demonstrating that the doped models retain the direct band gaps. Furthermore, due to the structural distortion induced by the doping atom, as shown in Figure 1 and Figure 2, the corresponding band gap also changes. The calculated band gaps of C−ZnS, Si−ZnS, Ge−ZnS, Sn−ZnS, and Pb−ZnS are 2.51, 2.91, 2.95, 2.91, and 2.86 eV, respectively. It is clear that the band gaps of the doped structures change little in comparison to the undoped ZnS, with the exception of C−ZnS. In addition, the intermediate states located at the band gap are observed in all of the doped models [36], which may prominently change the electronic and optical properties of the monolayer tetragonal ZnS.

Figure 5 shows the different charge densities for C−ZnS, Si−ZnS, Ge−ZnS, Sn−ZnS, and Pb−ZnS from both the top and side views, respectively, to identify electron transfer in the doped models. The yellow and blue areas correspond to the increasing and decreasing electron density, respectively. For C−ZnS, the S atoms near the C atom lose charge while the central C atom obtains charge, thus concentrating the charge on the C atom. In contrast to the case in C−ZnS, the different charge densities show a similar distribution for Si−ZnS, Ge−ZnS, Sn−ZnS, and Pb−ZnS. The charges on the central atoms (Si, Ge, Sn, and Pb) tend to transfer to the S atoms, concentrating the charges between central atoms and neighboring S atoms in the vicinity of the S atoms. Furthermore, the charge densities between the neighboring S atoms and Zn atoms increase, and they are further enhanced with the increasing metallicity of the doping atom. Interestingly, the doping atoms are always in a tetragonal environment, similar to the Zn atom, despite the diversity in charge density. 

The electron localization functions (ELF) of the ZnS monolayer and the associated doping models are shown in Figure 6. Usually, ELF = 0.5 corresponds to the metallic bond where electrons are totally delocalized, while ELF = 1 corresponds to the covalent bond where the electron is perfectly localized. For the ZnS monolayer, the values of ELF between the Zn and S atoms are relatively small, revealing that the Zn and S atoms tend to form ionic bonds. For C−ZnS, the values of ELF between C and S atoms are notably enhanced, and the ELF around the C atom is distributed uniformly, suggesting that C and S atoms tend to form σ bonds. As for Si−, Ge−, Sn−, and Pb−ZnS, the charge around the Si, Ge, Sn, and Pb atoms shows a prominent orientation deviating from the bonding direction, illustrating that the π bonds are formed in these cases. This diversity between doping elements is probably due to the difference in chemical bonds.

Figure 7 plots the projected density of state (PDOS) of the ZnS monolayer, C−ZnS, Si−ZnS, Ge−ZnS, Sn−ZnS, and Pb−ZnS. For the PDOS of the ZnS monolayer, the valence bands near the VBM are predominantly composed of the p states of Zn and S, indicating that Zn and S atoms tend to form p-p bonds despite their tetragonal structure. The CBM is ascribed to the s states of Zn, which originate from the outer−shell electron due to the high energies. When doping C into the ZnS monolayer, the s states of C and p states of S are generated above the valence band of the ZnS monolayer, and the p states of C are observed below the conduction band of the ZnS monolayer, resulting in a smaller band gap. The s states of the C and p states of S near the VBM form an sp^3^ hybrid interaction, resulting in the C−centered tetrahedron. The PDOS of monolayer Si−ZnS, Ge−ZnS, Sn−ZnS, and Pb−ZnS present similar electronic structures. The valence band structure is composed of two parts: one is the contribution from the Zn and S atoms resembling the ZnS monolayer, and the other is ascribed to the contribution from S and the doping atoms (Si, Ge, Sn, and Pb), in which the s states of the doping atoms and the p states of S atoms probably form the sp^3^ hybrid bonds. The conduction band structures of these models are similar to that of the ZnS monolayer, in which the CBM mainly comprises the s state of the Zn atom. Compared to the PDOS of the ZnS monolayer, it is clear that the group−ⅣA elements doping to the ZnS monolayer usually reduce the band structure by generating sp^3^ hybrid states in the band structure with higher energies. The prominent variance is probably due to the two extra valence electrons in the group−ⅣA elements, which vary the bonding nature in doped the ZnS monolayer.

Figure 8a,b shows the real (ε_1_) and imaginary (ε_2_) parts of the calculated dielectric function for the ZnS monolayer and the doped models. The ε_1_ of the doped models is larger than that of the ZnS monolayer when the energy is smaller than 2.1 eV, while they are smaller than that of the ZnS monolayer in the energy range of 2.1−6.5 eV. Beyond the scale, they are in good agreement. For the ε_2_, the values of doped models are larger than that of the ZnS monolayer in 1.2−6.1 eV, while they are smaller than that of ZnS in 6.1−7.0 eV. The values of these models are very consistent when the energy is above 7.0 eV. The optical absorption α(ω) of the doped ZnS models are shown in Figure 8c, which are obtained from ε_1_ and imaginary ε_2_ by the following formula:(2)$$\alpha \left(\omega \right)=\frac{\sqrt{2}\omega }{c}{\left[\sqrt{\varepsilon_{1}^{2}+\varepsilon_{2}^{2}}-\varepsilon_{1}\right]}^{\frac{1}{2}}$$

The absorption spectrum of the ZnS monolayer is predominantly above the 3.1 eV, suggesting that the ZnS monolayer has good absorption ability in the ultraviolet (UV) region. When doping group−ⅣA elements into the ZnS monolayer, the doped models still possess excellent absorption ability in UV region. Furthermore, a small absorption peak is observed in the vicinity of 2.0 eV for all of the doped models, suggesting that the group−ⅣA element doping to the ZnS monolayer could improve the absorption ability in the visible spectrum. In the doped models, the absorption peak of C−ZnS is largest in the visible region, indicating that C doping into the ZnS monolayer can effectively enhance the visible absorption of the ZnS monolayer.

## 4. Conclusions

In summary, by using a first-principles calculation, we have studied the electronic and optical properties of the ZnS monolayer after group−ⅣA element (C, Si, Ge, Sn, and Pb) doping. The results reveal that the formation energy of C−ZnS is the smallest, while those of Si−ZnS, Ge−ZnS, Sn−ZnS, and Pb−ZnS gradually decrease as the metallicities of the doping atoms increase. The doping atoms tend to form the sp^3^ hybrid bonds with neighboring S atoms. The doped models also exhibit a direct band gap. Unlike those of C−ZnS, the band gaps of the other doped models change little. Moreover, intermediate states are observed in the band gap, which indicates that the doped structures can exhibit an improved optical absorption ability in the visible spectrum. This finding provides us with an effective way to enhance the electronic and optical properties of materials with a tetragonal structure. Our research results enrich the understanding of the electronic and optical properties in the group−ⅣA element-doped ZnS monolayer, which can contribute to the development of applications in optoelectronic devices. 

## Figures and Tables

**Figure 1 nanomaterials-12-03898-f001:**
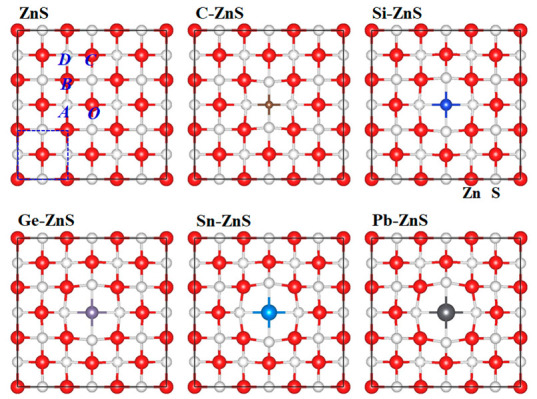
The supercell of the ZnS monolayer and the corresponding doped cases, with red and white balls representing Zn and S atoms, respectively.

**Figure 2 nanomaterials-12-03898-f002:**
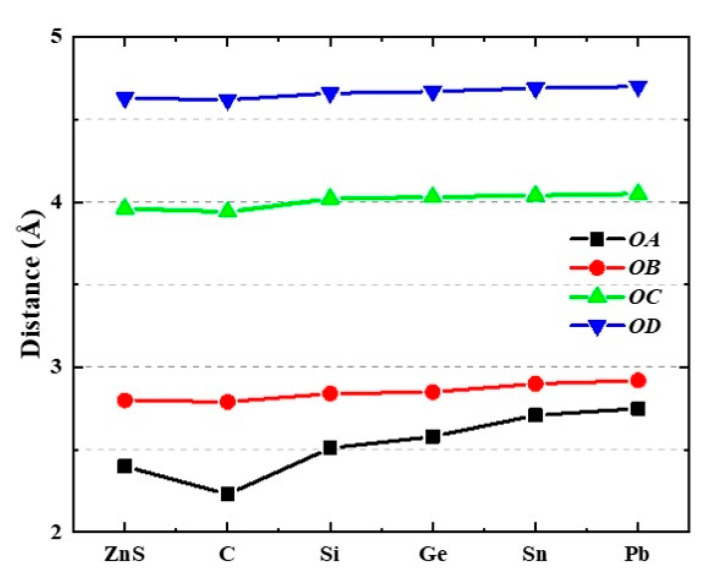
The interatomic distances from *O* atom to *A*, *B*, *C*, and *D* atoms, respectively.

**Figure 3 nanomaterials-12-03898-f003:**
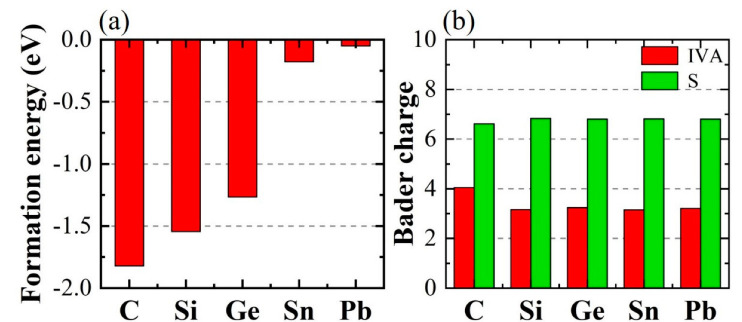
(**a**) Formation energies and (**b**) Bader charge of group−ⅣA element−doped the ZnS monolayer.

**Figure 4 nanomaterials-12-03898-f004:**
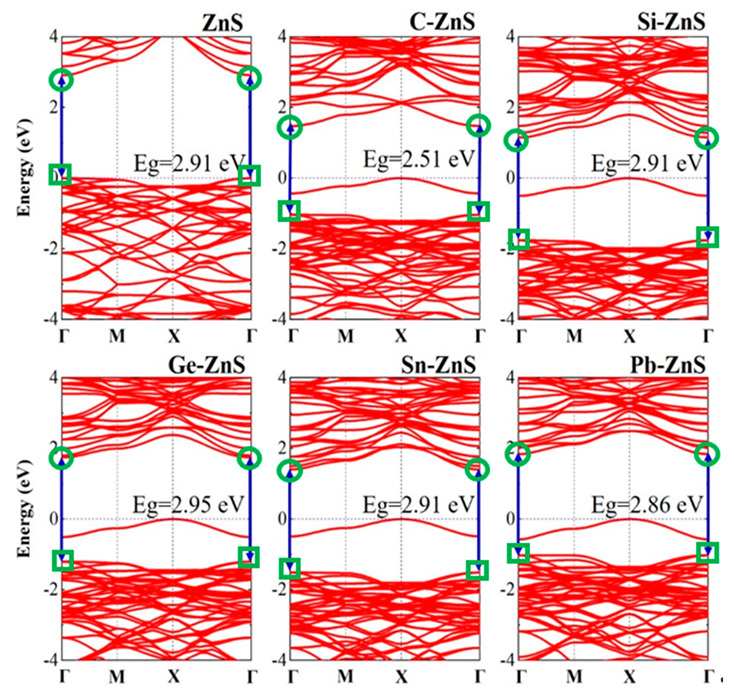
Band structures of the ZnS monolayer and the corresponding doped models. The circle and rectangle symbols represent the CBM and VBM, respectively.

**Figure 5 nanomaterials-12-03898-f005:**
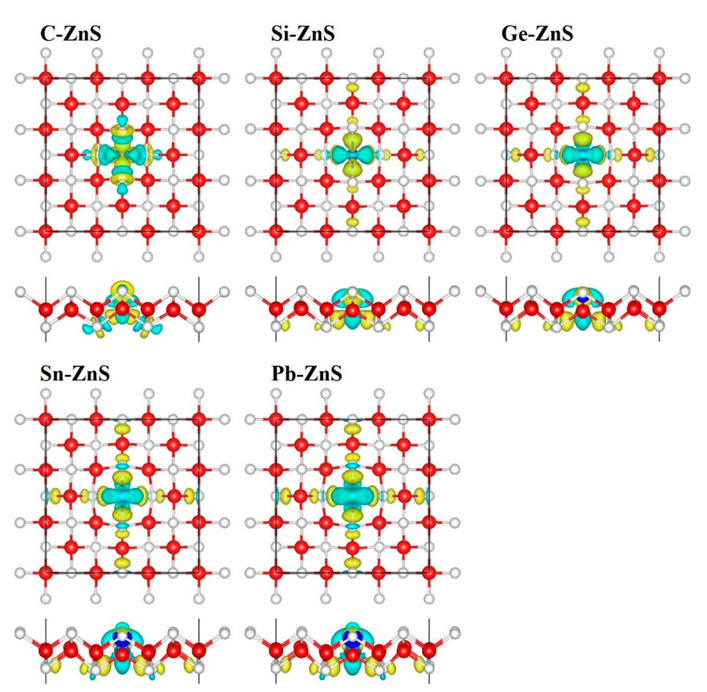
The difference in charge densities of monolayer C−ZnS, Si−ZnS, Ge−ZnS, Sn−ZnS, and Pb−ZnS, respectively, with an iso-surface of 0.002 Å^−2^.

**Figure 6 nanomaterials-12-03898-f006:**
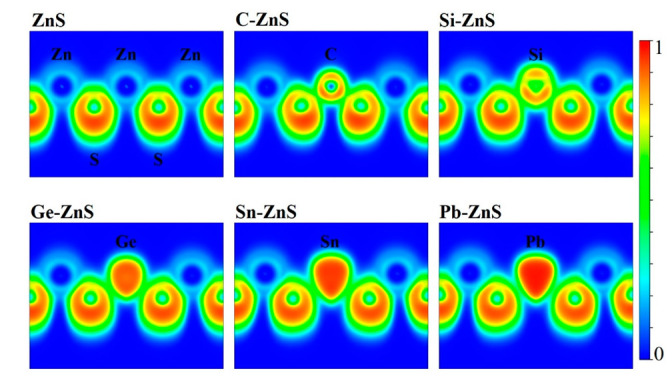
The electronic localization functions of ZnS and the doped models.

**Figure 7 nanomaterials-12-03898-f007:**
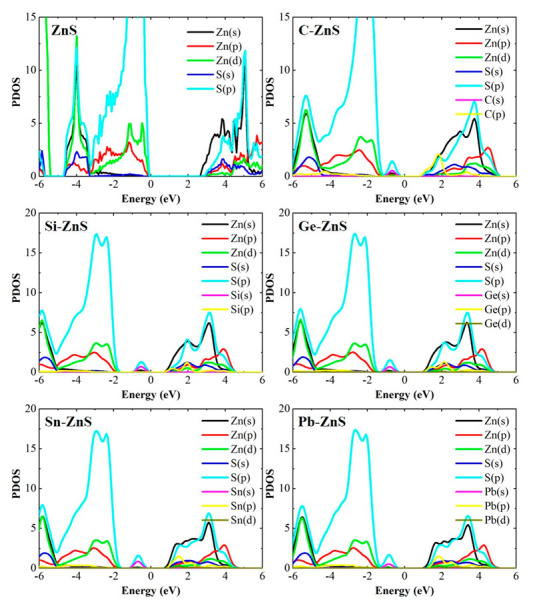
The PDOS of monolayer tetragonal ZnS and the corresponding doped models.

**Figure 8 nanomaterials-12-03898-f008:**
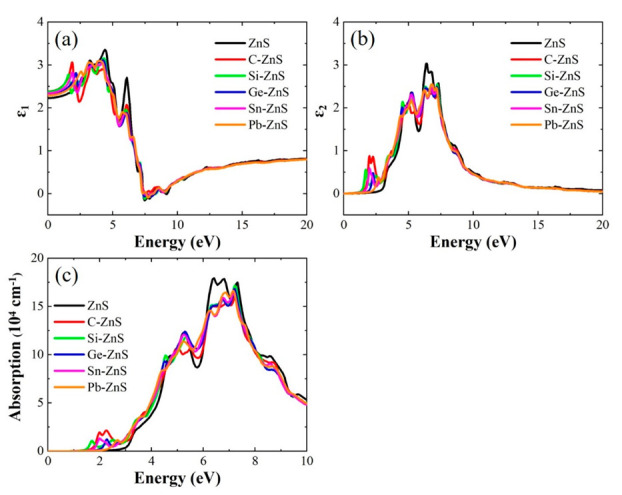
The (**a**) real and (**b**) imaginary parts of the dielectric function, and (**c**) optical absorption of ZnS monolayer and the doped cases.

## Data Availability

Not applicable.

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
