# Peer review of "Influence of Group-IVA Doping on Electronic and Optical Properties of ZnS Monolayer: A First-Principles Study"

_nanomaterials, 2022, doi:10.3390/nano12213898_

Round 1
Reviewer 1 Report
Using first-principles calculation, the authors studied the electronic and optical properties of monolayer ZnS doped with group-â…£A elements. The presentation of the results is clear and the conclusions are supported by calculations. As a side comment, I would recommend the authors to study the influence of lattice relaxation of the doped structures on the presented electronic and optical properties.
Reviewer 2 Report
The authors present a study of the electronic and optical properties of monolayer ZnS after group-IVA element doping by DFT. The paper is clearly written and the results are illustrated by well prepared figures.
Minor comment:
The paper could be better motivated and the authors should clearly indicate what is the new result they bring to the field.
Reviewer 3 Report
Comments:
In this work, the authors analyzed the influence of group-IVA doping on the electronic and optical properties of monolayer ZnS using first-principles calculations. They provided the formation energy analysis, and electronic and optical properties. They examine the associated doping structures in 2D ZnS for application in optoelectronic devices.
The results look reasonable. From a computational point of view, this is an interesting and systematic study using various state-of-the-art computational methods and tools.
I recommend the publication in Nanomaterials if the authors can address the following points in order to further improve the quality of the manuscript:
-After doping group-IV A element, the local atomic distortions are appeared. It is suggested to provide the amount of the charge transfer between the doped atom and its first nearest neighbor atoms. Moreover, a deep bond analysis (sigma, pi bonds) around doping atoms can be studied by electron localization function (ELF).
-Is the magnetic moment of the system changed after doping group-IV A element in the system. If yes, it is suggested to include the effect of magnetism on the electronic structure. For example, they can calculate the spin-polarized PBE electronic band structures.
-It is suggested to also include real and especially imaginary parts of the calculated dielectric function in Fig. 7.
